# Procrastination Predicts Future Internet Use Disorders in Adolescents but Not Vice Versa: Results from a 12-Month Longitudinal Study

**DOI:** 10.3390/healthcare11091274

**Published:** 2023-04-29

**Authors:** Julia Lardinoix, Ina Neumann, Lutz Wartberg, Katajun Lindenberg

**Affiliations:** 1Department of Child and Adolescent Psychotherapy, Institute of Psychology, Goethe University Frankfurt, 60486 Frankfurt, Germany; 2Department of Psychology, Faculty of Human Sciences, MSH Medical School Hamburg, 20457 Hamburg, Germany

**Keywords:** internet use disorders, procrastination, risk factors, adolescents, longitudinal study

## Abstract

Background: Use of Internet applications is often associated with postponing real-life obligations. Previous cross-sectional studies have reported a positive association between procrastination and Internet use disorders (IUDs). Further, procrastination is included in theoretical models explaining the development of IUDs. However, little is known about the temporal relationship; thus, longitudinal studies identifying the specific predictive direction are required. Methods: Using a cross-lagged panel design, a total of 240 students who were at high risk for IUDs, aged between 12 and 18 years (*M* = 15.17, *SD* = 1.66) were assessed at baseline (t1) and reassessed one year later (t2) with standardized measures for IUDs and procrastination. Results: Our results showed that IUD symptom severity at t2 was significantly predicted both by IUD symptom severity at t1 and procrastination at t1. However, procrastination at t2 was predicted only by procrastination at t1 but not by IUD symptom severity at t1. Conclusions: We could not find a reciprocal relationship between procrastination and IUD symptom severity. Procrastination appears to be a causal risk factor for IUDs, but not a consequence. As a modifiable risk factor, procrastination is a promising starting point for preventive and therapeutic interventions for IUDs.

## 1. Introduction

Internet applications provide various entertainment possibilities (e.g., gaming, streaming, social networks, and communication), which are available ubiquitously and are easily accessible in daily life. The COVID-19 pandemic has further expanded the relevance and influence of Internet usage. This can be observed in a considerable increase in time spent online, especially in adolescents [1,2,3,4]. Studies analyzing the time spent online by 12–19-year-old German adolescents report a daily use of 241–258 min compared to 205 min a day before the COVID-19 pandemic [5]. Increased daily time spent online is accompanied by a higher risk of developing Internet use disorders (IUDs) [6,7,8]. The prevalence of IUDs in German adolescents is currently estimated to be in the range of 3–6% [9,10,11,12]. Globally, IUD prevalence is approximately 6–7%, as reported in meta-analyses [13,14]. Adolescents and young adults are at a higher risk for developing IUDs than other age groups [15]. This increased vulnerability to developing IUDs can be attributed not only to the high amount of time spent online but also to the ongoing development of executive functions. Adolescents have a lower ability to control their actions because their executive functions are still developing [16], which makes independent control over the use of Internet applications more challenging at younger ages.

Thus, identifying potential risk factors for developing IUDs in adolescents is an important effort of IUD research to tailor preventive interventions. Use of Internet applications is often associated with postponing real-life obligations [17,18]. Therefore, for IUD research, the relationship with procrastination appears to be of particular interest. Procrastination has been significantly associated with IUDs in various studies, e.g., [19,20,21], and is additionally included in different explanatory models of IUDs, e.g., [22,23,24]. However, longitudinal studies, as well as studies identifying the specific predictive direction, are still lacking. Such studies are needed in order to gain a better understanding of the empirical relationship between these phenomena. To date, it is unclear whether procrastination is a cause, a consequence, or an associated symptom of IUDs.

### 1.1. Internet Use Disorders

IUDs are complex phenomena, which have recently been partially included as behavioral addictions in the ICD-11 [25]. The definition, especially a constant term (synonymous terms used in literature include “pathological Internet use”, “compulsive Internet use”, and “Internet addiction“), and the delimitation to other behavioral addictions are widely discussed in the extant literature.

In general, IUD can be seen as an umbrella term that includes different subtypes of online behavioral addictions [26]. IUD subsumes all gaming and nongaming Internet-related disorders (e.g., gaming disorder, social media disorder, pornography use disorder, and shopping disorder), which include the use of different digital media devices. Playing video games offline is also included in IUDs. As streaming disorder has not been sufficiently investigated so far, it is not officially listed as a subtype of IUD. Especially relevant for adolescents are three subtypes of IUD: gaming disorder, social media disorder and streaming disorder [27,28]. In this paper, we use the term IUD to describe the addictive use of games (online and offline), social media, and streaming in adolescents.

IUD can generally be defined as an increasingly excessive Internet use characterized by high intensity and duration, impaired control over usage, and negative consequences in psychosocial areas of life [22].

Through the addition of the newly included section of “behavioral addictions” in the latest version of the International Classification of Diseases (ICD-11), IUD has become increasingly important in daily clinical practice. Thereby, the WHO recognizes that not only substances but also behaviors can become addictive.

Gaming Disorder is the first IUD that has been classified as a new, independent diagnosis in this new ICD-11 section. In the DSM-5, Internet Gaming Disorder has only been included in section III as an emerging condition that requires further research [29]. Gaming Disorder manifests with three symptoms:(1)“Impaired control over gaming (e.g., onset, frequency, intensity, duration, termination, context)(2)Increasing priority given to gaming to the extent that gaming takes precedence over other life interests and daily activities(3)Continuation or escalation of gaming despite the occurrence of negative consequences”

Furthermore, according to the ICD-11, Gaming Disorder should lead to “…significant distress or impairment in personal, family, social, educational, occupational, or other important areas of functioning” [25] (p. 553).

Nongaming IUDs have not yet been included as independent diagnoses in the ICD-11 but can also be included in the “behavioral addictions” section, as they show similar characteristics to Gaming Disorder. According to research recommendations, Social Media Disorder can be classified as an “other specific behavioral addiction”, whereas Streaming Disorder can be classified as an “unspecified behavioral addiction” [26].

One of the main reasons why adolescents spend increasing amounts of time online is that Internet applications are highly rewarding, entertaining and motivating. Internet applications offer instant gratification through positive reinforcement (e.g., quick success, various stimuli, and flow experience), especially games that include specific mechanisms aiming to maximize playing time. At the same time, Internet applications offer distractions from negative feelings in real life as well as less attractive tasks and obligations [22,30,31,32,33,34].

### 1.2. Procrastination

Procrastination—the voluntary delay in starting or completing an intended course of action despite the expectation of a negative outcome for the delay [35]—is a widespread phenomenon [36,37,38,39]. The prevalence of procrastination has been rising during the past decades and is expected to rise further in the future [35]. Procrastination usually occurs when activities are regarded as unpleasant, tiresome, or challenging [40]. Procrastinators often choose short-term rewards instead of obtaining long-term gains [41].

Fundamentally, procrastination is interpreted as the result of deficient self-regulation, as reported in both quantitative and qualitative studies [42,43]. Procrastination goes along with many negative consequences for the individual, including health and mental health problems and social and occupational difficulties [44,45,46,47,48].

The reasons for procrastination and its behavioral manifestation are diverse. Research has suggested internal feeling states (e.g., fatigue), task characteristics [35], and attractive distractions in the personal environment (e.g., smartphones [49]) as triggers of procrastination, which can manifest in different behaviors [50]. In several studies [17,18,51,52] it has been reported that Internet applications, such as social media, or binge-watching television series are often used for procrastination.

### 1.3. Associations between Internet Use Disorder and Procrastination

Theoretically, there are multiple connections between IUDs and procrastination. Both IUDs and procrastination are characterized by deficient levels of self-regulation [22,30,42,43]. Adolescents with IUDs are unable to control their Internet use, whereas procrastinators have difficulties controlling their behavior while carrying out a task. Further, specific Internet applications (e.g., video games or social media) are entertaining, rewarding, and distracting [22,30,32,33,34]. Procrastinators favor short-term rewards and have the tendency to use digital media as a well available temptation in their immediate environment [41,49].

The association between IUDs and procrastination has been considered in the context of theoretical models of IUD, which assume that procrastination is a risk factor for IUDs (e.g., Brand et al.’s I-PACE model [22,30]; Davis’s cognitive–behavioral model [23]; the PROTECT etiology model [24]). Building on the I-PACE model [22,30], the PROTECT etiology model indicates a link between IUDs and procrastination. The PROTECT etiology model assumes that IUDs are associated with maladaptive coping in the context of regulating negative emotions. Three main areas have been identified as trigger mechanisms for negative effects: (1) motivational problems or boredom susceptibility, (2) performance anxiety that is often associated with procrastination of unpleasant tasks, and (3) social anxiety, often paired with a lack of social skills [24]. According to cognitive–behavioral theories, cognitive distortions (e.g., negative attribution style and overgeneralization) lead to negative evaluations in these three areas, which results in negative emotions. These negative emotions can be avoided by using Internet applications [24]. Thus, using Internet applications is used as a maladaptive coping strategy to compensate for negative emotions. This coping strategy works in the short term by causing positive emotions and gratification through Internet use. However, it is accompanied by negative long-term consequences in real life (e.g., poorer performance at school and more conflicts with friends and family), which causes more negative emotions. To regulate these negative emotions, the individual will most likely once again use Internet applications, which results in a vicious circle [24]. The PROTECT etiology model further includes specific behavioral, cognitive, and emotional therapeutic interventions to modify this maladaptive coping strategy. 

When evaluating the effectiveness of the intervention, Lindenberg et al. [53] found a significantly greater decrease in procrastination in the PROTECT intervention group compared to the control group with no intervention. Further, the PROTECT intervention group showed a significantly greater reduction in IUD symptom severity [53]. According to Davis’s cognitive–behavioral model [23], maladaptive cognitions favor the development and the maintenance of IUDs. He considers procrastination to be a central dimension of IUDs, as it is both a significant factor in the development and the maintenance of IUDs [23]. IUDs often manifest in the tendency of individuals to put off their responsibilities. This deficient time management can significantly impair daily functioning, as postponing important tasks and responsibilities leads to increased pressure on the individual [23].

However, the reverse—that IUDs affect procrastination—might theoretically also be the case. Students who are not in control of their own gaming and Internet use spend more time online and playing video games. This could lead to a shift in the precious resource of time, which is lacking elsewhere (for example, in the academic context), which in turn leads to poorer academic and cognitive performance [54,55,56,57]. Thus, students with lower self-control and problematic Internet use might have a greater tendency to avoid unpleasant activities and duties (e.g., schoolwork) [18,51].

A growing number of studies have empirically investigated associations between IUDs and procrastination. Previous cross-sectional studies have shown that procrastination is statistically significantly related to IUDs in adolescents [19,20,21] and adults [58,59,60,61,62]. In a cross-sectional study, Kindt et al. [21] investigated school-related risk factors for IUDs in students. They identified procrastination as an associated risk factor for IUDs [21].

Further studies have investigated specific forms of Internet usage and procrastination in children, adolescents, and young adults. A significant association has also been found between procrastination and specific Internet usage, such as problematic mobile phone use [50,63,64] or problematic social media use [64,65]. In an experience sampling study, Reinecke and Hofmann [66] found that media use conflicted with important goals and obligations in 60% of the sampled media use occurrences. In another study, one of the most common motives for students to engage in social media was to get away from pressures and responsibilities [67]. In a cross-lagged, longitudinal study, Hong et al. [63] found that procrastination preceded problematic mobile phone use one year later, but they found no stable reverse prediction. Using the mobile phone problem use scale, they could only investigate Internet usage related to mobile phones, but not IUDs in general. IUDs are independent of the devices used and cannot be investigated in general through mobile phone use. With regard to the addition of the newly included section of “behavioral addictions” in the latest version of the International Classification of Diseases [25], it is even more important to investigate not only problematic mobile phone use, but also the clinically relevant IUD diagnosis. Hinsch and Sheldon [51] showed in two intervention studies that reductions in Internet use resulted in significant decreases in procrastination behavior. Their findings indicate that a reverse relationship could also be assumed.

### 1.4. The Present Study

Although theoretical models include procrastination as an important risk factor for IUDs, empirical support is still scarce. As illustrated above, previous studies investigating the association of IUDs and procrastination have been typically based on cross-sectional designs. Further, empirical evidence to explain the temporal relationship between IUDs and procrastination is still missing.

In the present study, we investigated the predictive direction of IUD symptom severity and procrastination in a longitudinal perspective in adolescents. Thereby, we wanted to identify whether procrastination is a causal risk factor of IUD symptom severity or a consequence of the IUD. We applied a cross-lagged panel design in a longitudinal perspective, including two measurements over the course of one year (t1 = baseline, t2 = 12-month follow-up), to clarify the temporal relationship between IUDs and procrastination.

Following the theories and studies described previously, we hypothesized that IUD symptom severity and procrastination have a reciprocal relationship in adolescents.

## 2. Materials and Methods

### 2.1. Procedure

Data for the present study were collected via the PROTECT study (ClinicalTrials.gov: NCT02907658), a longitudinal, indicated prevention study. The goal of the PROTECT study was to investigate the effectiveness of a school-based preventive measure against IUDs in a randomized controlled design; however, only data from the untreated control group were used in this analysis. Ethical approval was obtained by the research ethics committee of the University of Education Heidelberg (Az.: 7741.35-13). Participants and parents signed a written informed consent before participating in the study. Data collection was conducted between October 2015 and September 2018 by trained psychologists in fifteen schools in three federal states in Germany (Baden-Wuerttemberg, Hessen, and Rhineland-Palatinate) during regular school hours. Data for the present investigation were collected in two assessments (t1 = baseline; t2 = 12-month follow-up).

### 2.2. Participants

In the present study, we included a total of 240 students (130 girls, or 54.2% of the sample, and 110 boys, or 45.8%) at elevated risk for developing IUDs, who did not receive any intervention during the 12-month observation period. All participants were identified with an initial screening. We used the compulsive Internet use scale (CIUS) for screening [68,69], for which excellent psychometric properties had been shown in a representative sample of German adolescents [70], and included participants with a CIUS score of 20 or higher. Participants were between 12 and 18 years old (M = 15.17; SD = 1.66). The majority (75%) attended a school with a high educational level (“Gymnasium”, *n* = 180), 17% attended a school with an intermediate educational level (“Realschule”, *n* = 40), and 8% attended a school with a low educational level (“Hauptschule/Werkrealschule”, *n* = 20).

### 2.3. Materials

Within our longitudinal study, participants completed different self-report questionnaires, including data on sociodemographics, procrastination, and IUD symptoms. All questionnaires, except the one on sociodemographics (only assessed at baseline, t1), were assessed both at baseline (t1) and one year later at the 12-month follow-up (t2).

#### 2.3.1. Internet Use Disorder—Modified Version of the German Video Game Dependency Scale

To assess IUD symptom severity at t1 and t2, we used a modified version of the German video game dependency scale (“Computerspielabhängigkeitsskala für Jugendliche”; CSAS-J [71]). The original questionnaire measures gaming disorders, using all 9 diagnostic criteria of the DSM-5, which are (1) preoccupation, (2) withdrawal, (3) tolerance, (4) loss of control, (5) giving up other activities, (6) continuation, (7) deception, (8) escape, and (9) negative consequences [29], using two items per criterion (18 in total). Items were modified to cover not only gaming disorder but also IUDs in adolescents in general (e.g., “Even when I am not gaming/online, I think about gaming/going online” and “I feel that I can no longer control the time I spend on video games/the Internet”). All items were rated on a 4-point Likert scale (0 = *not right* to 3 = *exactly right*). Besides categorical classification, a sum score can be computed and interpreted using grade- and age-specific norms. In the present study, we used the sum score to measure the IUD. The reliability coefficients (Cronbach’s α) for the CSAS-J were 0.81 at t1 and 0.86 at t2 in the surveyed sample. Further information on the psychometric properties of the CSAS-J can be found in Rehbein et al. [57,71].

#### 2.3.2. Procrastination—German General Procrastination Scale

We applied the German general procrastination scale [72] to asses procrastination. The instrument consists of 3 subscales (18 items in total): (1) procrastination (7 items, e.g., “I postpone important tasks until the last moment.”), (2) aversion to tasks (6 items, e.g., “I feel uncomfortable when I need to begin working on important tasks.”), and (3) preference for alternatives (5 items, e.g., “Before I start with an important task, I prefer dealing with other things first.”). All items can be rated on a 7-point Likert scale (1 = *never* to 7 = *always*). For each subscale, mean scores can be computed using gender-specific norms. In the present study, procrastination was measured by computing the sum score of all three scales. The reliability coefficients (Cronbach’s α) for the German general procrastination scale were 0.94 at t1 and 0.95 at t2 in the sample studied. Psychometric properties of the German general procrastination scale were reported in Höcker et al. [72].

### 2.4. Data Analyses

The aim of our study was to investigate the predictive direction of IUD symptom severity and procrastination in a longitudinal perspective (12-month follow-up) in adolescents. Therefore, we first ran bivariate correlation analyses. For our main analyses, we calculated cross-lagged panel models in a longitudinal perspective to identify whether procrastination is a risk factor of IUD symptom severity, or whether IUD symptom severity causes procrastination. (In the bivariate and in the multivariate analyses, we did not use bootstrap analysis to determine the 95% confidence intervals). The statistical analyses were conducted with IBM SPSS Statistics Version 27 for Windows (IBM, 2020, New York, NY, USA) and Mplus (to calculate the structural equation models). The dependent variables in the structural equation models were procrastination (at t2) and IUD (at t2). As explanatory variables in the structural equation models, we used gender and age of the adolescent as well as procrastination (at t1) and IUD (at t1). Because we examined two measurement time points, we used a cross-lagged panel model (CLPM) instead of a random-intercept cross-lagged panel model (RI-CLPM), which requires at least three measurements [73]. First, we calculated a structural equation model without restrictions (a full path model with zero degrees of freedom). To assess the global goodness-of-fit of the model, we used as fit indices the comparative fit index (CFI) and the Tucker–Lewis index (TLI), as well as the root-mean-square error of approximation (RMSEA) and the standardized root-mean-square residual (SRMR). To be able to determine the global goodness-of-fit indices for the model (RMSEA, SRMR, CFI, and TLI), we fixed the regressions coefficients to be equal for both sexes and calculated a second model. According to Schermelleh-Engel et al. [74] the cut-off values for a good model fit were RMSEA ≤ 0.05, SRMR ≤ 0.05, CFI ≥ 0.97, and TLI ≥ 0.97.

## 3. Results

### 3.1. Descriptive Analyses

The mean values of IUD symptoms were *M* = 12.90 (*SD* = 7.02) at t1 and *M* = 9.39 (*SD* = 6.86) at t2, measured using a modified version of the CSAS-J [71]). The average values for procrastination were *M* = 69.84 (*SD* = 20.42) at t1 and *M* = 68.32 (*SD* = 21.19) at t2 (data were collected with the German general procrastination scale [72]).

### 3.2. Correlation Analyses

In the bivariate correlation analyses, we observed statistically significant associations between procrastination (at t1) and procrastination (at t2), IUD (at t1), and IUD (at t2, see Table 1). IUD (at t1) was related to procrastination (at t1) and IUD (at t2). Furthermore, procrastination (at t2) was associated with IUD (at t2), and age of the participant with procrastination (at t1), procrastination (at t2), and IUD (at t1, see also Table 1).

### 3.3. Structural Equation Model without Restrictions

In the structural equation model without restrictions, we found stronger procrastination (at t1) to be predictive of procrastination one year later (at t2, see Table 2). The model explained one-third of the variance (33 percent) of procrastination at t2. More symptoms of IUD (at t1) and stronger procrastination (at t1) were predictors of IUD a year later (at t2). The model explained 25 percent of the variance in Internet use disorder at t2 (see right column of Table 2).

### 3.4. Structural Equation Model with Restrictions

In the SEM, with regression coefficients restricted to be equal for both sexes, we obtained a good global goodness-of-fit for the model (RMSEA = 0.00, SRMR = 0.06, CFI = 1.00 and TLI = 1.01). Again, procrastination (t2) was predicted by stronger procrastination at t1 (see Table 3, explained variance was 0.37 for girls and 0.27 for boys), whereas more IUD symptoms (at t1) and stronger procrastination (at t1) were predictive of IUD (at t2, explained variance was 0.24 for females and 0.27 for males).

## 4. Discussion

The aim of this longitudinal study was to investigate the predictive direction of IUD symptom severity and procrastination in adolescents, testing the theoretical models that include procrastination as a risk factor for development of IUDs, as well as the reverse theory assuming that problematic Internet use results in higher procrastination. The cross-lagged panel model showed that earlier procrastination (t1) appears to be an important risk factor for IUD symptom severity one year later (t2). The reverse prediction could not be found.

We found that IUD symptom severity (t2) was predicted by procrastination and IUD symptom severity at t1. This finding is consistent with previous cross-sectional research [21]. As a result of deficient self-regulation, procrastination is characterized by postponing tasks [35]. Procrastinators have the tendency to use Internet applications as a well available temptations in their environment instead of carrying out a task [18]. Many Internet applications are highly entertaining and motivating and offer short-term rewards, which is why adolescents, especially those with the tendency to procrastinate, spend a lot of time using digital media. Therefore, it seems quite reasonable that procrastination precedes IUD symptom severity.

Contrary to our expectations, we could not find a reciprocal relationship between procrastination and IUD symptom severity. Regarding the reverse prediction, we found that later procrastination (t2) is predicted by procrastination at t1 but not by IUD symptom severity at t1. Our finding is in line with Hong et al. [63], who analyzed the longitudinal relationship between procrastination and problematic mobile phone use and could only find procrastination as a predictor for problematic phone use but no stable reverse prediction. However, unlike our study, they only examined problematic smartphone use and not IUDs in general. Internet applications include mechanisms that aim to maximize the usage time and offer distractions from less attractive tasks and obligations at the same time. We thus expected IUD symptom severity to predict procrastination one year later because we assumed that adolescents postpone their tasks because they prefer spending a lot of time using Internet applications. Our data show that procrastination is a causal risk factor that predicts future IUDs. This relationship is not bidirectional, i.e., procrastination is not a consequence of the IUD.

Our longitudinal findings go beyond previous findings on the association of procrastination and IUDs. Previous studies have only considered the association between procrastination and IUDs in a cross-sectional perspective. In contrast, we applied a cross-lagged, longitudinal perspective to analyze the specific predictive direction of procrastination and IUDs. Thus, we strengthened previous findings [21] and provided longitudinal evidence to support procrastination preceding IUD symptom severity.

In another study, Hong et al. [63] applied a cross-lagged, longitudinal perspective in which they considered procrastination and problematic mobile phone use. By using the mobile phone problem use scale, they could only investigate Internet usage related to mobile phones, but not IUDs in general. Regarding the addition of the newly included section of “behavioral addictions” in the latest version of the International Classification of Diseases [25], it is even more important to not only consider problematic mobile phone use but also the clinically relevant IUD diagnosis, which is independent of the device used.

Although our results give a clear answer to the temporal relationship between procrastination and IUD symptom severity in adolescents, some aspects need to be considered when interpreting the results. First, our sample included only adolescents at elevated risk for IUDs, which limits the generalizability to individuals not at elevated risk. Further, using a modified version of the German video game dependency scale (CSAS-J; [71]) to assess IUD symptom severity, it was not possible to differentiate between the subtypes of IUD. Additionally, because we used a questionnaire instead of a diagnostic interview, we can only identify procrastination as a risk factor for IUD symptoms, not IUD diagnoses. To generalize our findings, it would also be interesting to investigate the relationship between procrastination and IUDs based on a sample of adolescents not at elevated risk for IUDs in future research. Further, more research is needed to understand the relationship between procrastination and the subtypes of IUD to see whether the predictive direction differs between procrastination and the distinct subtypes. It might also be of interest to investigate the mechanisms underlying the relationship between procrastination and IUDs. Furthermore, it is very likely that some other, additional, aspects influence the relationship between IUDs and procrastination (e.g., as moderators or mediators). In previous research, for example, stress, sleep quality, and relationship with the parents were mentioned [75]. For the relationship with the parents (e.g., on the aspect of adolescent autonomy), initial, recently published empirical findings on IUDs are now also available [76]. Difficulties in emotion regulation are a relevant factor for both procrastination and IUDs [55,77,78]. Therefore, the role of emotion regulation in the relationship between procrastination and IUDs should be addressed. In future studies, it would be important to consider all of these aspects together to further describe and understand the relationships between IUDs and procrastination. From a methodological perspective, more measurement time points could be considered in future surveys to utilize random-intercept cross-lagged panel models. Additionally, 95% confidence intervals could be calculated with bootstrapping.

## 5. Conclusions

The results of our cross-lagged, longitudinal study enhance the knowledge on the association between procrastination and IUDs. Procrastination significantly predicted an increase in symptom severity of IUD at the 12-month follow-up, but not vice versa. Thus, procrastination could be identified as a significant, causal risk factor of IUDs. Because procrastination is a modifiable risk factor, it is a promising starting point for practical implications, such as preventive and therapeutic interventions for IUDs, for example, the PROTECT intervention [24]. PROTECT is a cognitive–behavioral therapy intervention for adolescents aged 12–18 years who spend a high amount of time using Internet applications. PROTECT was designed to include classical cognitive–behavioral therapy interventions, such as cognitive restructuring, behavior activation, problem-solving training, and techniques for emotion regulation. The intervention consists of four modules, which refer to difficulties associated with IUDs: (1) boredom and motivational problems, (2) procrastination and performance anxiety, (3) social anxiety and friendship, and (4) emotion regulation [24]. In the context of this study, module two is particularly relevant. Adolescents are taught useful alternatives to using digital media as well as strategies to change negative thought patterns and unpleasant emotional states in the context of procrastination [24]. A recent RCT has shown that besides the effects of the PROTECT intervention on the primary outcome, significant incremental effects on procrastination over 12 months were found [53]. Thus, the reduction in procrastination plays an important role in preventing IUDs.

## Figures and Tables

**Table 1 healthcare-11-01274-t001:** Correlation matrix for the variables.

Variable	1	2	3	4	5	6
(1) Gender ^a^	–					
(2) Age	0.05	–				
(3) Procrastination (t1)	0.09	0.17 *	–			
(4) Procrastination (t2)	0.08	0.17 *	0.56 **	–		
(5) Internet Use Disorder (t1)	−0.09	−0.17 *	0.24 **	0.05	–	
(6) Internet Use Disorder (t2)	−0.01	−0.09	0.28 **	0.38 **	0.44 **	–

Note. ^a^ Coding: 1 = male, 2 = female. ** *p* < 0.01; * *p* < 0.05.

**Table 2 healthcare-11-01274-t002:** Results of the structural equation model without restrictions.

Variable	Procrastination (t2) Standardized Beta Coefficients	Internet Use Disorder (t2) Standardized Beta Coefficients
Procrastination (t1)	0.57 ***	0.21 **
Internet use disorder (t1)	−0.07	0.39 ***
Gender ^a^	0.04	0.01
Age	0.06	−0.09
*R* ^2^	0.33	0.25

Note. ^a^ Coding: 1 = male, 2 = female. *** *p* < 0.001; ** *p* < 0.01

**Table 3 healthcare-11-01274-t003:** Results of the structural equation model with restrictions.

Variable	Procrastination (t2) Standardized Beta Coefficients		Internet Use Disorder (t2) Standardized Beta Coefficients	
	Girls	Boys	Girls	Boys
Procrastination (t1)	0.61 ***	0.52 ***	0.23 **	0.19 **
Internet use disorder (t1)	−0.06	−0.07	0.37 ***	0.42 ***
Age	0.06	0.06	−0.09	−0.08
*R* ^2^	0.37	0.27	0.24	0.27

Note. *** *p* < 0.001; ** *p* < 0.01. Model fit: RMSEA = 0.00; SRMR = 0.06; CFI = 1.00; TLI = 1.01.

## Data Availability

The raw data supporting the conclusions of this article will be made available by the corresponding author upon request.

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
