# Peer review of "Procrastination Predicts Future Internet Use Disorders in Adolescents but Not Vice Versa: Results from a 12-Month Longitudinal Study"

_healthcare, 2023, doi:10.3390/healthcare11091274_

Round 1

Reviewer 1 Report

The article addresses a relevant, and unfortunately increasingly important issue, the IUD. The chosen focal point is a nice addition to the understanding of the phenomenon.

The paper operates with fresh literature, out of 53 only 3 is not for the 21st century, what is more 14 is from the last 3 years.

Both IUD and procrastination are introduced in depth and their realisation is described based on previous academic findings.

The chosen methodology is adequate, what is more the longitudinal nature of the research make the results more trustworthy.

The results are presented in a clear and understandable manner. The results are in line with the data and are validated against the relevant international literature.

Conclusions are rather short. Please, reflect on how these results can be utilised in practice! I would also invite the authors to disclose the limitations and the future plans as well.

Author Response

Reviewer #1:

Point #1: The article addresses a relevant, and unfortunately increasingly important issue, the IUD. The chosen focal point is a nice addition to the understanding of the phenomenon. The paper operates with fresh literature, out of 53 only 3 is not for the 21st century, what is more 14 is from the last 3 years. Both IUD and procrastination are introduced in depth and their realisation is described based on previous academic findings. The chosen methodology is adequate, what is more the longitudinal nature of the research make the results more trustworthy. The results are presented in a clear and understandable manner. The results are in line with the data and are validated against the relevant international literature.

Answer #1: We thank the reviewer for the insightful feedback and hope that the expansion of the conclusions is to the reviewer’s satisfaction.

Point #2: Conclusions are rather short. Please, reflect on how these results can be utilised in practice! I would also invite the authors to disclose the limitations and the future plans as well.

Answer #2: We expanded the conclusions by adding the PROTECT intervention to give an example for how to use the results in practice: “Since procrastination is a modifiable risk factor, it is a promising starting point for practical implications, such as preventive and therapeutic interventions for IUD, for example the PROTECT intervention (Lindenberg et al., 2020). PROTECT is a cognitive behavioral therapy intervention for adolescents aged 12 – 18 years, who spend a high amount of time using Internet applications. PROTECT was designed to include classical cognitive behavioral therapy interventions such as cognitive restructuring, behavior activation, problem-solving training and techniques for emotion regulation. The intervention consists of four modules, which refer to difficulties associated with IUD: (1) boredom and motivational problems, (2) procrastination and performance anxiety, (3) social anxiety and friendship, and (4) emotion regulation (Lindenberg et al., 2020). In the context of this study, module 2 is particularly relevant. Adolescents are taught useful alternatives to using digital media as well as strategies to change negative thought patterns and unpleasant emotional states in the context of procrastination (Lindenberg et al., 2020). A recent RCT has shown that besides the effects of the PROTECT intervention on the primary outcome, significant incremental effects on procrastination were found (Lindenberg et al., 2022). Thus, the reduction of procrastination plays an important role in preventing IUD.”

We discussed the limitations of our study in the discussion and added some more future research plans as follows: “Although our results give a clear answer on the temporal relationship between procrastination and IUD symptom severity in adolescents, some aspects need to be considered when interpreting the results. Firstly, our sample included only adolescents at elevated risk of IUD which limits the generalizability to individuals without elevated risk. Further, using a modified version of the German Video Game Dependency Scale (CSAS-J) to assess IUD symptom severity, it was not possible to differentiate be-tween the subtypes of IUD. Additionally, since we used a questionnaire instead of a diagnostic interview, we can only identify procrastination as a risk factor for IUD symptoms, not IUD-diagnoses. To generalize our findings, it would be interesting to also investigate the relationship between procrastination and IUD based on a sample of adolescents without elevated risk of IUD in future research. Further, more research is needed to understand the relationship between procrastination and the subtypes of IUD to see if the predictive direction differs between procrastination and the distinct subtypes. It might also be of interest to investigate the mechanisms underlying the relationship between procrastination and IUD. It is very likely that additionally some other aspects influence the relationships between IUD and procrastination (e.g. as moderators or mediators). In previous research, for example, stress, sleep quality, and relationship with the parents were mentioned (Reinecke et al., 2018); for the relationship with the parents (e.g. on the aspect of adolescent autonomy), initial, recently published empirical findings on IUD are now also available (Wartberg et al., 2022). Difficulties in emotion regulation are a relevant factor for both procrastination and IUD (e.g., Gentile et al., 2011; Sirois et al., 2019; Tice & Bratslavsky, 2000). Therefore, the role of emotion regulation in the relationship between procrastination and IUD should be addressed. In future studies, it would be important to consider at all of these aspects together to further describe and understand the relationships between IUD and procrastination.”

Reviewer 2 Report

Dear authors,

the paper I reviewed aims to present a longitudinal study exploring the predictive direction between procrastination and Internet Use Disorders in adolescents. While the paper is interesting and original, major edits have to be made for accepting the contribution. I’ve tried to highlight the key points which should be reviewed. These mainly include an in-depth description of the introduction section and, above all, a more careful and precise treatment of the method and results sections.

In detail:

MAIN TEXT:

I want to begin with a general comment about Introduction section: the focus here is on adolescents because the study sample consists of adolescents, but the entire section lacks a full discussion of adolescent age. I think it would be useful to either include a dedicated paragraph related to adolescence or devote more space in the first paragraph to this age stage so that subsequent paragraphs can connect to this by describing the data that studied the two variables of your interest (IUD and procrastination) in relation to adolescence.

Pp. 1 (lines 28-34): “Internet applications provide various entertainment possibilities (e.g., gaming, streaming, social networks, communication), which are available ubiquitously and easily accessible in daily life. The COVID-19 pandemic has further expanded the relevance and influence of Internet usage. This can be observed in a considerable increase in time spent online, especially in adolescents. Studies analyzing time spent online of 12–19 year old German adolescents report a daily use of 241–258 minutes compared to 205 minutes a day before the COVID-19 pandemic [1]”. More recent papers need to be found.

Pp. 1 (lines 34-35): “Increased daily time spent online is accompanied by a higher risk of developing Internet Use Disorders (IUD) [2]” More recent papers need to be found.

Pp. 1 (lines 41-43): “Procrastination has been significantly associated with IUD in various studies [11–13] and is additionally included in different explanatory models of IUD [14–16].” Literature reviews could be included if they exist.

Pp. 2, (lines 49-51): “IUDs are a complex phenomenon, which are included in in the disorder category of behavioral addictions[17]”. Considering that this phenomenon is still under studied and is partially present in ICD-11 only but not in DSM 5 (it is only present in section III) more space needs to be devoted otherwise the term "disorder category" remains unclear.

Pp. 2, (lines 66-69): “Through the addition of the newly included section of „behavioral addictions” in the latest version of the International Classification of Diseases (ICD-11), IUD has become increasingly important in daily clinical practice. Thereby, WHO recognizes that not only substances but also excessive behaviors (e.g., gaming) can become addictive”. It is important to better explain what is meant by "excessive behaviors." I am unclear about the use of the adjective "excessive". Gaming does not necessarily become addictive.

Pp. 2 (lines 70-72): “Gaming Disorder” is the first IUD which has been classified as new, independent diagnosis into this new ICD-11 section (besides the inclusion of “Internet Gaming Disorder” in DSM-5; [20])”. It is important to clarify that in DSM 5 this phenomenon is not included as a “recognized disorder” because it is in section III pending future studies.

Pp. 2 (lines 79-80): “Furthermore according to ICD-11, gaming should lead to “...significant distress or impairment in personal, family, social, educational, occupational, or other important areas of functioning” [21]”. The page number per exact citation should be entered.

Pp. 2 (lines 88-93): “One of the main reasons why adolescents increasingly spent a lot of time online is that Internet applications are highly rewarding, entertaining and motivating. Internet applications offer instant gratification through positive reinforcement (e.g., quick success, various stimuli, flow experience). Especially games include specific mechanisms aiming to maximize playing time. At the same time, Internet applications offer distractions from negative feelings in real life as well as less attractive tasks and obligations”. Recent papers need to be found to support these claims.

Pp. 2 (lines 96-98): “Procrastination - the voluntary delay to starting or completing an intended course of action despite the expectation of a negative outcome for the delay - [22] is a widespread phenomenon”. Recent papers need to be found to support this claim.

Pp. 3 (lines 14-19): “Theoretically, there are multiple connections between IUD and procrastination. Both IUD and procrastination are characterized by deficient levels of self-regulation. Adolescents with IUD are unable to control their Internet use, while procrastinators have difficulties controlling their behavior while carrying out a task. Further, specific Internet applications (e.g., video games or social media) are entertaining, rewarding and distracting. Procrastinators favor short-term rewards and have the tendency to use digital media as a well available temptation in their immediate environment”. Recent papers need to be found to support these claims.

Pp. 4 (lines 52-58): “However, the reverse direction - the effect of IUD on procrastination - might theoretically also be the case. Students who are not in control of their own gaming and Internet use, spend more time online and playing video games. This could lead to a shift in the precious resource of time, which is lacking elsewhere, for example in the academic context, which in turn leads to poorer academic and cognitive performance. Thus, students with lower self-control and problematic Internet use might have a greater tendency to avoid unpleasant activities and duties (e.g., schoolwork)”. Papers need to be found to support these claims (if available) and the reverse direction hypothesis.

About Materials and Methods, it is important to include internal consistency values for each measure. Secondly, have the measures used been validated in your country? If yes, validation studies should be included in the references. Thirdly, was the significance of the effects determined using the boostrap method? If yes how many samples?

Pp. 6 (lines 267-268): “To be able to determine the global goodness-of-fit indices for the model (RMSEA, SRMR, CFI and TLI), we fixed the regressions coefficients to be equal for both sexes and calculated a second model”. Cut-off values used to evaluate the goodness of fit indexes of the model must be entered.

Method could benefit from a more in-depth description of the goodness-of-fit indexes. Secondly, by what criterion did you choose some criteria – RMSEA, SRMR, CFI and TLI – and not others? A reference needs to be included. Finally, have you checked the normality of the data?

About Results, I think it may be useful to analyze the data using a random-intercept cross-lagged panel model (RI-CLPM) to get results that are less prone to bias. See Etherson et al., 2022 "Feelings of not Mattering and Depressive Symptoms From a Temporal Perspective: A Comparison of the Cross-Lagged Panel Model and Random-Intercept Cross-Lagged Panel Model".

I think it might be helpful to include a figure that clearly shows the cross-lagged model. it might be helpful for readers.

About Discussion, more space could be devoted to the clinical and research implications. For example, could be interesting to discuss about the emotional components of procrastination (e.g., difficulties in emotion regulation). Overall, the section could benefit from more discussion of adolescent-related implications and research future directions.

Author Response

Reviewer #2:

Point #1: I want to begin with a general comment about Introduction section: the focus here is on adolescents because the study sample consists of adolescents, but the entire section lacks a full discussion of adolescent age. I think it would be useful to either include a dedicated paragraph related to adolescence or devote more space in the first paragraph to this age stage so that subsequent paragraphs can connect to this by describing the data that studied the two variables of your interest (IUD and procrastination) in relation to adolescence.

Answer #1: We appreciate the helpful comments of the reviewer on revising some aspects of our paper. We included the reviewer’s comments and hope that the changes are to the reviewer’s satisfaction.

We added the following paragraph related to adolescents to the introduction to put the focus on the investigated age group: Adolescents are at a higher risk for developing IUD than other age groups (Przepiroka et al., 2019). This increased vulnerability to developing IUD can be attributed not only to the high amount of time spent online but also the ongoing development of executive functions. Adolescents have a lower ability to control their actions because their executive functions are still developing (McCelland & Cameron, 2011), which makes independent control over the use of Internet applications more challenging at younger ages.”

Point #2: Pp. 1 (lines 28-34): “Internet applications provide various entertainment possibilities (e.g., gaming, streaming, social networks, communication), which are available ubiquitously and easily accessible in daily life. The COVID-19 pandemic has further expanded the relevance and influence of Internet usage. This can be observed in a considerable increase in time spent online, especially in adolescents. Studies analyzing time spent online of 12–19 year old German adolescents report a daily use of 241–258 minutes compared to 205 minutes a day before the COVID-19 pandemic [1]”. More recent papers need to be found.

Answer #2: We added the following papers to support our claims:

  1. Trott, R. Driscoll, E. Irlado, and S. Pardhan, “Changes and correlates of screen time in adults and children during the COVID-19 pandemic: A systematic review and meta-analysis,” EClinicalMedicine, vol. 48, p. 101452, 2022, doi: 10.1016/j.eclinm.2022.101452.
  2. M. Nagata et al., “Screen Time Use Among US Adolescents During the COVID-19 Pandemic: Findings From the Adolescent Brain Cognitive Development (ABCD) Study,” JAMA pediatrics, vol. 176, no. 1, pp. 94–96, 2022, doi: 10.1001/jamapediatrics.2021.4334.
  3. Lampert, K. Thiel, and B. Güngör, “Mediennutzung und Schule zur Zeit des ersten Lockdowns während der Covid-19-Pandemie 2020: Ergebnisse einer Online-Befragung von 10- bis 18-Jährigen in Deutschland,” 2021.
  4. A. Kovacs et al., “Physical activity, screen time and the COVID-19 school closures in Europe - An observational study in 10 countries,” European journal of sport science, vol. 22, no. 7, pp. 1094–1103, 2022, doi: 10.1080/17461391.2021.1897166.

Point #3: Pp. 1 (lines 34-35): “Increased daily time spent online is accompanied by a higher risk of developing Internet Use Disorders (IUD) [2]” More recent papers need to be found.

Answer #3: We added the following papers to support our claims:

  1. Neumann and K. Lindenberg, “Internetnutzungsstörungen unter deutschen Jugendlichen vor und während der COVID-19-Pandemie,” Kindheit und Entwicklung, vol. 31, no. 4, pp. 193–199, 2022, doi: 10.1026/0942-5403/a000390.
  2. Wartberg, L. Kriston, K. Kegel, and R. Thomasius, “Adaptation and Psychometric Evaluation of the Young Diagnostic Questionnaire (YDQ) for Parental Assessment of Adolescent Problematic Internet Use,” Journal of behavioral addictions, vol. 5, no. 2, pp. 311–317, 2016, doi: 10.1556/2006.5.2016.049.

Point #4: Pp. 1 (lines 41-43): “Procrastination has been significantly associated with IUD in various studies [11–13] and is additionally included in different explanatory models of IUD [14–16].” Literature reviews could be included if they exist.

Answer #4: We could not find a relevant literature review to include here.

Point #5: Pp. 2, (lines 49-51): “IUDs are a complex phenomenon, which are included in in the disorder category of behavioral addictions[17]”. Considering that this phenomenon is still under studied and is partially present in ICD-11 only but not in DSM 5 (it is only present in section III) more space needs to be devoted otherwise the term "disorder category" remains unclear.

Answer #5: We changed the sentence to the following, to make clear, that IUDs are partially present in ICD-11: “IUDs are a complex phenomenon, which have recently been partially included as behavioral addictions in ICD-11 (WHO, 2018).”

Point #6: Pp. 2, (lines 66-69): “Through the addition of the newly included section of „behavioral addictions” in the latest version of the International Classification of Diseases (ICD-11), IUD has become increasingly important in daily clinical practice. Thereby, WHO recognizes that not only substances but also excessive behaviors (e.g., gaming) can become addictive”. It is important to better explain what is meant by "excessive behaviors." I am unclear about the use of the adjective "excessive". Gaming does not necessarily become addictive.

Answer #6: As we only wanted to point out that non-substance related addictions have been recognized by WHO, we changed the sentence to the following: “Thereby, WHO recognizes that not only substances, but also behaviors can become addictive.”

Point #7: Pp. 2 (lines 70-72): “Gaming Disorder” is the first IUD which has been classified as new, independent diagnosis into this new ICD-11 section (besides the inclusion of “Internet Gaming Disorder” in DSM-5; [20])”. It is important to clarify that in DSM 5 this phenomenon is not included as a “recognized disorder” because it is in section III pending future studies.

Answer #7: To clarify that “Internet Gaming Disorder” is not included as a “recognized disorder”, we changed the sentence as follows: “Gaming Disorder” is the first IUD which has been classified as new, independent diagnosis into this new ICD-11 section. In DSM-5, “Internet Gaming Disorder” has only been included in section III as an emerging condition that requires further research (APA, 2013).”

Point #8: Pp. 2 (lines 79-80): “Furthermore according to ICD-11, gaming should lead to “...significant distress or impairment in personal, family, social, educational, occupational, or other important areas of functioning” [21]”. The page number per exact citation should be entered.

Answer #8: We added the page number to the exact citation: “Furthermore according to ICD-11, gaming should lead to “...significant distress or impairment in personal, family, social, educational, occupational, or other important areas of functioning (WHO, 2018, p. 553).”

Point #9: Pp. 2 (lines 88-93): “One of the main reasons why adolescents increasingly spent a lot of time online is that Internet applications are highly rewarding, entertaining and motivating. Internet applications offer instant gratification through positive reinforcement (e.g., quick success, various stimuli, flow experience). Especially games include specific mechanisms aiming to maximize playing time. At the same time, Internet applications offer distractions from negative feelings in real life as well as less attractive tasks and obligations”. Recent papers need to be found to support these claims.

Answer #9: We added the following papers to support our claims:

  1. Brand et al., “The Interaction of Person-Affect-Cognition-Execution (I-PACE) model for addictive behaviors: Update, generalization to addictive behaviors beyond internet-use disorders, and specification of the process character of addictive behaviors,” Neuroscience and biobehavioral reviews, vol. 104, pp. 1–10, 2019, doi: 10.1016/j.neubiorev.2019.06.032.
  2. Thalemann, K. Wölfling, and S. M. Grüsser, “Specific cue reactivity on computer game-related cues in exces-sive gamers,” Behavioral neuroscience, vol. 121, no. 3, pp. 614–618, 2007, doi: 10.1037/0735-7044.121.3.614.
  3. Stavropoulos, M. D. Griffiths, T. L. Burleigh, D. J. Kuss, Y. Y. Doh, and R. Gomez, “Flow on the internet: A longitudinal study of Internet addiction symptoms during adolescence,” Behaviour & Information Technology, vol. 37, pp. 159–172, 2018, doi: 10.1080/0144929X.2018.1424937.
  4. J. Kuss and M. D. Griffiths, “Internet and gaming addiction: a systematic literature review of neuroimaging studies,” Brain sciences, vol. 2, no. 3, pp. 347–374, 2012, doi: 10.3390/brainsci2030347.
  5. D. Griffiths, “IS THE BUYING OF LOOT BOXES IN VIDEO GAMES A FORM OF GAMBLING OR GAMING?,” Gaming Law Review, vol. 22, no. 1, pp. 52–54, 2018, doi: 10.1089/glr2.2018.2216.
  6. Brand, K. S. Young, C. Laier, K. Wölfling, and M. N. Potenza, “Integrating psychological and neurobiological considerations regarding the development and maintenance of specific Internet-use disorders: An Interaction of Person-Affect-Cognition-Execution (I-PACE) model,” Neuroscience and biobehavioral reviews, vol. 71, pp. 252–266, 2016, doi: 10.1016/j.neubiorev.2016.08.033.

Point #10: Pp. 2 (lines 96-98): “Procrastination - the voluntary delay to starting or completing an intended course of action despite the expectation of a negative outcome for the delay - [22] is a widespread phenomenon”. Recent papers need to be found to support this claim.

Answer #10: We added the following papers to support our claims:

  1. J. Solomon and E. D. Rothblum, “Academic procrastination: Frequency and cognitive-behavioral correlates,” Journal of Counseling Psychology, vol. 31, pp. 503–509, 1984, doi: 10.1037/0022-0167.31.4.503.
  2. C. Schouwenburg, “Procrastination in Academic Settings: General Introduction,” in Counseling the procrasti-nator in academic settings, Washington, DC, US: American Psychological Association, 2004, pp. 3–17.
  3. U. Özer, A. Demir, and J. R. Ferrari, “Exploring Academic Procrastination Among Turkish Students: Possible Gender Differences in Prevalence and Reasons,” The Journal of Social Psychology, vol. 149, no. 2, pp. 241–257, 2009, doi: 10.3200/SOCP.149.2.241-257.
  4. Mohammadi bytamar, S. Zenoozian, M. Dadashi, O. Saed, A. Hemmat, and G. Mohammadi, “Prevalence of Academic Procrastination and Its Association with Metacognitive Beliefs in Zanjan University of Medical Sci-ences,” J Med Educ Dev, vol. 10, 2018, doi: 10.29252/edcj.10.27.84.

Point #11: Pp. 3 (lines 14-19): “Theoretically, there are multiple connections between IUD and procrastination. Both IUD and procrastination are characterized by deficient levels of self-regulation. Adolescents with IUD are unable to control their Internet use, while procrastinators have difficulties controlling their behavior while carrying out a task. Further, specific Internet applications (e.g., video games or social media) are entertaining, rewarding and distracting. Procrastinators favor short-term rewards and have the tendency to use digital media as a well available temptation in their immediate environment”. Recent papers need to be found to support these claims.

Answer #11: We added the following papers to support our claims:

  1. Brand et al., “The Interaction of Person-Affect-Cognition-Execution (I-PACE) model for addictive behaviors: Update, generalization to addictive behaviors beyond internet-use disorders, and specification of the process character of addictive behaviors,” Neuroscience and biobehavioral reviews, vol. 104, pp. 1–10, 2019, doi: 10.1016/j.neubiorev.2019.06.032.
  2. Thalemann, K. Wölfling, and S. M. Grüsser, “Specific cue reactivity on computer game-related cues in exces-sive gamers,” Behavioral neuroscience, vol. 121, no. 3, pp. 614–618, 2007, doi: 10.1037/0735-7044.121.3.614.
  3. Stavropoulos, M. D. Griffiths, T. L. Burleigh, D. J. Kuss, Y. Y. Doh, and R. Gomez, “Flow on the internet: A longitudinal study of Internet addiction symptoms during adolescence,” Behaviour & Information Technology, vol. 37, pp. 159–172, 2018, doi: 10.1080/0144929X.2018.1424937.
  4. J. Kuss and M. D. Griffiths, “Internet and gaming addiction: a systematic literature review of neuroimaging studies,” Brain sciences, vol. 2, no. 3, pp. 347–374, 2012, doi: 10.3390/brainsci2030347.
  5. Brand, K. S. Young, C. Laier, K. Wölfling, and M. N. Potenza, “Integrating psychological and neurobiological considerations regarding the development and maintenance of specific Internet-use disorders: An Interaction of Person-Affect-Cognition-Execution (I-PACE) model,” Neuroscience and biobehavioral reviews, vol. 71, pp. 252–266, 2016, doi: 10.1016/j.neubiorev.2016.08.033.
  6. Steel and K. B. Klingsieck, “Academic procrastination: Psychological antecedents revisited,” Australian Psychologist, vol. 51, no. 1, pp. 36–46, 2016, doi: 10.1111/ap.12173.

Wendelien van Eerde and Katrin B. Klingsieck, “Overcoming procrastination? A meta-analysis of intervention studies,” Educational Research Review, 2018.

  1. Tice and R. Baumeister, “Longitudinal Study of Procrastination, Performance, Stress, and Health: The Costs and Benefits of Dawdling,” Psychological Science - PSYCHOL SCI, vol. 8, pp. 454–458, 1997, doi: 10.1111/j.1467-9280.1997.tb00460.x.
  2. Steel, F. Svartdal, T. Thundiyil, and T. Brothen, “Examining Procrastination Across Multiple Goal Stages: A Longitudinal Study of Temporal Motivation Theory,” Frontiers in psychology, vol. 9, p. 327, 2018, doi: 10.3389/fpsyg.2018.00327.

Point #12: Pp. 4 (lines 52-58): “However, the reverse direction - the effect of IUD on procrastination - might theoretically also be the case. Students who are not in control of their own gaming and Internet use, spend more time online and playing video games. This could lead to a shift in the precious resource of time, which is lacking elsewhere, for example in the academic context, which in turn leads to poorer academic and cognitive performance. Thus, students with lower self-control and problematic Internet use might have a greater tendency to avoid unpleasant activities and duties (e.g., schoolwork)”. Papers need to be found to support these claims (if available) and the reverse direction hypothesis.

Answer #12: We added the following papers to support our claims:

  1. Rehbein, S. Kliem, D. Baier, T. Mößle, and N. M. Petry, “Prevalence of Internet gaming disorder in German adolescents: diagnostic contribution of the nine DSM-5 criteria in a state-wide representative sample,” Addiction (Abingdon, England), vol. 110, no. 5, pp. 842–851, 2015, doi: 10.1111/add.12849.
  2. Rehbein, M. Kleimann, and T. Mössle, “Prevalence and risk factors of video game dependency in adolescence: results of a German nationwide survey,” Cyberpsychology, behavior and social networking, vol. 13, no. 3, pp. 269–277, 2010, doi: 10.1089/cyber.2009.0227.
  3. A. Gentile et al., “Pathological video game use among youths: a two-year longitudinal study,” Pediatrics, vol. 127, no. 2, e319-29, 2011, doi: 10.1542/peds.2010-1353.
  4. S. Brunborg, R. A. Mentzoni, and L. R. Frøyland, “Is video gaming, or video game addiction, associated with depression, academic achievement, heavy episodic drinking, or conduct problems?,” Journal of behavioral addictions, vol. 3, no. 1, pp. 27–32, 2014, doi: 10.1556/JBA.3.2014.002.
  5. Meier, L. Reinecke, and C. Meltzer, "Facebocrastination"? Predictors of using Facebook for procrastination and its effects on students' well-being,” Computers in Human Behavior, vol. 64, pp. 65–76, 2016, doi: 10.1016/j.chb.2016.06.011.
  6. Hinsch and K. M. Sheldon, “The impact of frequent social Internet consumption: Increased procrastination and lower life satisfaction,” J. Consumer Behav., vol. 12, no. 6, pp. 496–505, 2013, doi: 10.1002/cb.1453.

Point #13: About Materials and Methods, it is important to include internal consistency values for each measure. Secondly, have the measures used been validated in your country? If yes, validation studies should be included in the references. Thirdly, was the significance of the effects determined using the boostrap method? If yes how many samples?

Answer #13: As recommended by Reviewer 2, we have calculated internal consistency values for the utilized measures (the range was between 0.81 to 0.95) and inserted them in the revised manuscript. References on the psychometric properties of the CSAS-J and the German general procrastination scale were also added in the Measures section of the revised manuscript. We calculated correlation analyses and cross-lagged panel models (SEMs), but did not use any bootstrap methods in the data analyses of the present study.

Point #14: Pp. 6 (lines 267-268): “To be able to determine the global goodness-of-fit indices for the model (RMSEA, SRMR, CFI and TLI), we fixed the regressions coefficients to be equal for both sexes and calculated a second model”. Cut-off values used to evaluate the goodness of fit indexes of the model must be entered.

Answer #14:  As recommended by Reviewer 2, we have included cut-off values for the fit indices in the revised manuscript (in the section "Data analyses").

Point #15: Method could benefit from a more in-depth description of the goodness-of-fit indexes. Secondly, by what criterion did you choose some criteria – RMSEA, SRMR, CFI and TLI – and not others? A reference needs to be included. Finally, have you checked the normality of the data?

Answer #15: We described the utilized fit indices a bit more detailed in the revised manuscript and a new reference was included (Schermelleh-Engel et al, 2003) in the bibliography. We selected the CFI, the TLI, and the RMSEA because they are sensitive to model misspecifications and do not depend on sample size (see Schermelleh-Engel et al, 2003). Because we used very established and robust procedures for statistical analysis, we had not checked the normal distribution of the data.

Point #16: About Results, I think it may be useful to analyze the data using a random-intercept cross-lagged panel model (RI-CLPM) to get results that are less prone to bias. See Etherson et al., 2022 "Feelings of not Mattering and Depressive Symptoms From a Temporal Perspective: A Comparison of the Cross-Lagged Panel Model and Random-Intercept Cross-Lagged Panel Model".

Answer #16: According to Usami's (2021) comparative article ("On the Differences between General Cross-Lagged Panel Model and Random-Intercept Cross-Lagged Panel Model: Interpretation of Cross-Lagged Parameters and Model Choice") [1], the "...RI-CLPM [Random-Intercept Cross-Lagged Panel Model] is identified if two or more variables have been measured at three or more time points, whereas the CLPM [Cross-Lagged Panel Model] requires only two time points" (p. 332). Therefore, we decided to calculate a Cross-Lagged Panel Model with the two measurment points in our study (t1 and t2).

Point #17: I think it might be helpful to include a figure that clearly shows the cross-lagged model. it might be helpful for readers.

Answer #17: We agree with Reviewer 2 that in complex modeling, graphical illustrations often improve understanding, but in our results only four numbers (Table 2) or eight numbers (Table 3) are relevant to understanding the key messages (though we find the presentation in two tables to be very clear).

Point #18: About Discussion, more space could be devoted to the clinical and research implications. For example, could be interesting to discuss about the emotional components of procrastination (e.g., difficulties in emotion regulation). Overall, the section could benefit from more discussion of adolescent-related implications and research future directions.

Answer #18: We changed and expanded the discussion, to devote more space to the clinical and research applications as follows: “To generalize our findings, it would be interesting to also investigate the relationship between procrastination and IUD based on a sample of adolescents without elevated risk of IUD in future research. Further, more research is needed to understand the relationship between procrastination and the subtypes of IUD to see if the predictive direction differs between procrastination and the distinct subtypes. It might also be of interest to investigate the mechanisms underlying the relationship between procrastination and IUD. It is very likely that additionally some other aspects influence the relationships between IUD and procrastination (e.g. as moderators or mediators). In previous research, for example, stress, sleep quality, and relationship with the parents were mentioned (Reinecke et al., 2018); for the relationship with the parents (e.g. on the aspect of adolescent autonomy), initial, recently published empirical findings on IUD are now also available (Wartberg et al., 2022). Difficulties in emotion regulation are a relevant factor for both procrastination and IUD (e.g., Gentile et al., 2011; Sirois et al., 2019; Tice & Bratslavsky, 2000). Therefore, the role of emotion regulation in the relationship between procrastination and IUD should be addressed. In future studies, it would be important to consider at all of these aspects together to further describe and understand the relationships between IUD and procrastination.”

References

  1. Usami S (2021) On the Differences between General Cross-Lagged Panel Model and Random-Intercept Cross-Lagged Panel Model: Interpretation of Cross-Lagged Parameters and Model Choice. Structural Equation Modeling: A Multidisciplinary Journal 28:331–344. https://doi.org/10.1080/10705511.2020.1821690

Round 2

Reviewer 2 Report

Dear authors,

the paper I reviewed aims to present a longitudinal study exploring the predictive direction between procrastination and Internet Use Disorders in adolescents. I would like to thank the authors for the edits they have made, but I believe that further minor revisions are needed before accepting the paper.

MAIN TEXT:

- It is necessary to include in both the method (Data analyses paragraph) and study’s limitations sections that the results do not present the use of 95% bootstrapped Cis.

- It is necessary to justify in the method section the use of cross-lagged panel model (CLPM) instead of random intercept cross-lagged panel model (RI-CLPM) by also adding a reference similarly to what was done in the response to my comment #16.

- I think may be useful to include effect size cutoff values for calculated models (see Orth et al., 2022 “Effect Size Guidelines for Cross-Lagged Effects”).

Author Response

Point #1: The paper I reviewed aims to present a longitudinal study exploring the predictive direction between procrastination and Internet Use Disorders in adolescents. I would like to thank the authors for the edits they have made, but I believe that further minor revisions are needed before accepting the paper.

It is necessary to include in both the method (Data analyses paragraph) and study’s limitations sections that the results do not present the use of 95% bootstrapped Cis.

Answer #1: As suggested by the Reviewer, we have added this aspect in the paragraph “Data analyses” as well as in the “limitations section” of the discussion.

Point #2: It is necessary to justify in the method section the use of cross-lagged panel model (CLPM) instead of random intercept cross-lagged panel model (RI-CLPM) by also adding a reference similarly to what was done in the response to my comment #16.

Answer #2: As suggested by the Reviewer, we have also added this aspect in the Data analyses paragraph and a new reference (Usami, 2021 or [73]) was added in the bibliography.

Point #3: I think may be useful to include effect size cutoff values for calculated models (see Orth et al., 2022 “Effect Size Guidelines for Cross-Lagged Effects”).

Answer #3: We read the article by Orth et al. (2022) with interest, but we are not completely convinced by the approach. In our opinion, instead of a general designation as "small, medium or large effect," it would be more useful to compare our results with other cross-lagged effects between IUD and procrastination (as soon as further studies are available, which is why we refrained from integrating general cut off values in the manuscript).

References

Orth, U., Meier, L. L., Bühler, J. L., Dapp, L. C., Krauss, S., Messerli, D., & Robins, R. W. (2022). Effect size guidelines for cross-lagged effects. Psychological Methods, doi: 10.1037/met0000499

Usami, S. “On the Differences between General Cross-Lagged Panel Model and Random-Intercept Cross-Lagged Panel Model: Interpretation of Cross-Lagged Parameters and Model Choice,” Structural Equation Modeling: A Multi-disciplinary Journal, vol. 28, no. 3, pp. 331–344, 2021